# Monitoring and Identification of Agricultural Crops through Multitemporal Analysis of Optical Images and Machine Learning Algorithms

**DOI:** 10.3390/s22166106

**Published:** 2022-08-16

**Authors:** José M. Espinosa-Herrera, Antonia Macedo-Cruz, Demetrio S. Fernández-Reynoso, Héctor Flores-Magdaleno, Yolanda M. Fernández-Ordoñez, Jesús Soria-Ruíz

**Affiliations:** 1Colegio de Postgraduados, Campus Montecillo, Carretera México-Texcoco, Km. 36.5, Montecillo, Texcoco 56230, Estado de México, Mexico; 2Sitio Experimental Metepec, Instituto Nacional de Investigaciones Forestales y Agropecuaria (INIFAP), Vial Adolfo López Mateos, Km. 4.5 Carretera Toluca Zitácuaro, Zinacantepec 51350, Estado de México, Mexico

**Keywords:** support vector machine, bagged trees, vegetation index, corn, bean, alfalfa

## Abstract

The information about where crops are distributed is useful for agri-environmental assessments, but is chiefly important for food security and agricultural policy managers. The quickness with which this information becomes available, especially over large areas, is important for decision makers. Methodologies have been proposed for the study of crops. Most of them require field survey for ground truth data and a single crop map is generated for the whole season at the end of the crop cycle and for the next crop cycle a new field survey is necessary. Here, we present models for recognizing maize (*Zea mays* L.), beans (*Phaseolus vulgaris* L.), and alfalfa (*Medicago sativa* L.) before the crop cycle ends without current-year field survey for ground truth data. The models were trained with an exhaustive field survey at plot level in a previous crop cycle. The field surveys begin since days before the emergence of crops to maturity. The algorithms used for classification were support vector machine (SVM) and bagged tree (BT), and the spectral information captured in the visible, red-edge, near infrared, and shortwave infrared regions bands of Sentinel 2 images was used. The models were validated within the next crop cycle each fifteen days before the mid-season. The overall accuracies range from 71.9% (38 days after the begin of cycle) to 87.5% (81 days after the begin cycle) and a kappa coefficient ranging from 0.53 at the beginning to 0.74 at mid-season

## 1. Introduction

Remote sensing has several potential applications in the agricultural field since it allows us to observe the distribution, area, and phenological states of crops. In this sense, Orynbaikyzy et al. [1] mention that the spectral characteristics of optical data are useful to determine the condition of the vegetation, the chlorophyll content, the water content, and the phenology of plants. However, achieving an accurate classification of crops can be difficult due to the phenological stages of crops among other factors. Several authors have reported spectral variations throughout phenological development and spectral similarities between crop classes [2,3]. In addition, development and cultivation patterns may vary within crops due to the production systems implemented by different producers [4], an issue that is particularly prevalent in Mexican agriculture.

The temporal changes in the crop phenology identified by satellite images monitoring during the agricultural cycle can help to achieve better classifications. Data from unmanned aerial vehicles and satellite images provide information for crop monitoring, but frequent data acquisition along the entire growth period of the crop at least during the key stages of growth and development is necessary [5]. In this sense, multitemporal analysis of images has provided greater precision than the analysis of data from a single image [2]. However, meteorological conditions do not always allow access to images useful for analysis due to the presence of clouds and haze in study areas. To mitigate this problem, the selection of images with a low percentage of clouds and high temporal resolution is needed. The use of unmanned aerial vehicles or drones can also encounter inconveniences and limitations, including costs associated with better spatial and temporal resolutions and geometric and radiometric corrections, which have been analyzed in precision agriculture studies [6].

Another crucial factor to consider when a study aims to obtain an accurate classification of crops is the selection of the appropriate recognition model. Among these models, nonparametric methods, which do not assume any distribution between classes, stand out. Support vector machines (SVM) are one type of nonparametric model and can achieve high classification accuracy with high-dimensional data, even if the size of the training dataset is small [7]. SVM were developed in the 1960s, but their popularity grew in the 1990s, when they were incorporated into the field of computational science. SVM has been shown to be one of the best supervised machine learning classifiers for a wide range of situations and is therefore considered a standard within the field of statistical and machine learning [8]. The SVM algorithm can perform classifications, regressions, and detect outliers. Although it was designed for binary classification, its use has been extended to cases of multiple classes, which is common in remote sensing [9]. Another method of machine learning for modelling potential characteristics is the bagged trees (BT) algorithm proposed by Breiman [10]. This technique is based on multiple trees and consists of taking subsets of data and training the decision trees repeatedly with replacement. The best classification is voted upon, and a general classification is obtained [11].

Based on the above considerations and with the objective of identifying crops at different dates and phenological stages, the changes in the reflectance of plots were monitored during an agricultural cycle through the multitemporal analysis of continuous optical images acquired by Sentinel-2. A methodology was proposed that considers samples or observations of the plots in different phenological stages within the agricultural cycle since crops are not planted on the same date due to variations in the production systems implemented by each producer. Each plot was visited as many times as possible, and visits were made to coincide with the passage of the satellite to achieve the best correlation between the state of the plot and the information captured by the satellite. Therefore, the characteristics of the training samples would reflect the truth in the field, mainly if the crop had already emerged. In total, 27 cloud-free images acquired between April and September 2019 were used for analysis. The features extracted from the satellite image was the reflectance mean of 20 m bands (B5, B6, B7, B8a, B11 and B12), 10 m bands (B2, B3 and B4) resampled to 20 m, and two vegetation indices, including the normalized difference vegetation index (NDVI) [12] and the weighted differences vegetation index (WDVI) [13]. This last is particularly important for determining whether crops have emerged or if a plot has already been harvested. Finally, to enable the consideration of crops in any of their phenological stages, the database was structured in such a way that each row or record was the crop in any of its phenological stages and its attributes or descriptors were the reflectance and vegetation indices at the current phenology stage plus the reflectance and vegetation indices of the same area over past satellite images in a fixed sequence. The models classify each crop according to the temporal profile of the descriptors during and before the end of crop cycle. The model was trained with data from 2019 and was tested in the next agricultural cycle (2020). The results were evaluated with records of the crops from the irrigation module using the kappa coefficient, with a good recognition of the crops.

The rest of the document is organized as follows. In Section 2, the materials and methods are described. To provide more context for the classification of the crops, a description of the crop pattern is presented. Furthermore, descriptions are given for the sampling method and sample size; selection of satellite images; construction of databases; and training of classification, validation, and testing models. In Section 3, the results of each of the trained classification models (SVM and BT) as well as a comparison of the two models are presented and discussed. The testing of the models in the subsequent cycle is also discussed in Section 3. The conclusions are presented in Section 4.

### 1.1. Related Works

With the launch of Earth observation satellites into space, crop mapping has represented a potential application. Table 1 shows various approaches for crop classification, highlighting the use of freely accessible images such as MODIS, Landsat, Sentinel and the use of machine learning classification methods. The first approaches were to obtain the annual crop map or crop cycle map using a single image [14,15] or a series of images [16,17], that cover an entire agricultural year or a crop cycle. In both cases only one map is generated for the whole analysis. The second approach is to map the crops in near real time over large areas [18,19] with the objective of generate crop maps before the end of the crop cycle. These approaches have limitations (among them, the time and cost of field survey). Authors such as Cai et al. [20] and Konduri et al. [21] have used multi-year analyzes to find growth patterns and use them in classification without the need of fields survey for ground truth observation.

### 1.2. Contributions of This Work

The proposed approach in this work allows for the mapping of crops throughout the agricultural cycle and not only on a specific date or at the end of the agricultural cycle.

The proposed methodology allows for obtaining classification models that can be used in the same area in subsequent agricultural cycles at any date within the agricultural cycle and could potentially be used even in nearby areas with the same crop distribution.

In optical images, the presence of clouds is a limitation to generate models, but the proposed methodology does not require a series of continuous images, since it can even be calibrated with a pair of cloud-free images.

The procedure is applicable for use in regions where agriculture is small and the average plot size is 1 hectare or less, since in North America there are few studies for smallholders agricultural land areas [25].

The disadvantage of the methodology is that extensive field survey is needed to train the models and an updated vector layer of plots is required to carry out the classification in subsequent agricultural cycles.

## 2. Materials and Methods

Classification with Sentinel 2 images time combinations and monitoring fields mainly consists of five steps, including (1) the construction of time-phenology of crop database; (2) the extraction of features [B2–B8A; B11–B12; Normalized Difference Vegetation Index (NDVI), Weighted Difference Vegetation Index (WDVI)] into a database; (3) constructing the database with the four combinations proposed; (4) obtaining a BT and SVM classification model based on the database; and (5) validating during the next agricultural cycle. The methodology is summarized in Figure 1 until the generation of the models and Figure 2 for validation on the next crop cycle.

### 2.1. Study Area

The study area was the Irrigation Module 05 (Tepatepec), which belongs to Irrigation District 003 Tula, Hidalgo in Mexico. It has an area of 5440 hectares distributed in 5244 parcels, located between the coordinates 99°06′00″ west longitude and 20°25′00″ north latitude, as shown in Figure 3.

The field survey covered all of the phenological stages of the crops, including the preparation of the land prior to sowing (bean and corn), during the 2019 Spring-Summer Crop Cycle (Crop cycle).

### 2.2. Crop Pattern

The Crop cycle begins at the beginning of April and ends at the end of September, and the main crops are corn (*Zea mays* L.), which accounted for 67.2% of the total crops; alfalfa (*Medicago sativa* L.), 22.5%; and bean (*Phaseolus vulgaris* L.), 5.9%. The remaining 4.4% of the total crops were other crops. Alfalfa cuts are variable and occur every eight weeks. The maize crop begins to be sown in April and is harvested at the end of September. Beans sowing begins in April, with harvesting in early July when the crop is in the grain filling stage. Table 2 shows the patterns of the main crops of the Tepatepec Irrigation Module.

### 2.3. Data Input and Processing

(a)Field survey data and frequency

To monitor crops throughout the crop cycle, the cluster sampling pattern was chosen [27] since it is the most economical and fastest approach; this sampling pattern can result in self-correlation problems [28], although the probability of this happening was low since the sampling unit was the parcel, with uniform and independent management of contiguous parcels.

Through stratified random sampling, 280 plot grouped into clusters within the 5 sections of the module were selected for monitoring (Figure 4). The total crops consisted of 154 plots of corn, 72 plots of alfalfa, and 54 plots of beans.

A total of 92 revisits were conducted in the Irrigation Module during the agricultural cycle. In each visit, the phenological stages was recorded at the plot level; in addition, photographs were taken of the plots visited. The frequency of plot monitoring was not uniform. On average, each parcel was visited 12 times, for a total of 3215 visits. Figure 5 shows the number of parcels visited per month in the study area during the 2019 crop cycle.

Of the total numbers of corn, bean, and alfalfa plots, 140 (50%) were randomly selected and used to train the classifier, with the remaining 140 (50%) plots used for validation.

(b)Sampling unit taken from satellite images

According to Congalton and Green [27], sampling units can consist of pixels, groups of pixels or polygons or groups of polygons. In this research, the sampling units consisted of groups of pixels or polygons, which were not less than nine pixels, or polygons, preferably of three rows by three columns [28]. However, the shape and orientation of a parcel does not always allow the selection of a square cluster. In some cases, clusters of 8 or 10 pixels were selected, as shown in Figure 6.

The sampling unit was digitized onto the Sentinel-2 images using free QGIS software.

The sample size was standardized due to the high difference in size of the plots and the cluster was drawn according to the best position within the plot as shown in the Figure 6. With the cluster layer and zone statistics tool in QGIS, the average reflectance of the 8 or 9 or 10 pixels in each of the analyzed bands was added to the attribute table of the shapefile (one field for each band).

(c)Satellite images used

Time series of satellite scenes were used to monitor and detect changes in the development of the crops. In total, 27 Sentinel-2 images were used, and the acquisition dates of the scenes ranged from April to September 2019.

Table 3 shows the dates of the cloud-free images used in the present investigation (in green) and the images with clouds (in red); the dates in red were not used in this investigation.

The Sentinel-2A and Sentinel-2B satellites have a temporal resolution of 5 days and spatial resolutions of 10 m (in four bands), 20 m (in nine bands) and 60 m (in three bands).

In the present study, we chose to use Sentinel-2 images with a spatial resolution of 20 m, since these images met the requirements for application in agriculture.

The bands used were B2 (490 nm), B3 (560 nm), B4 (665 nm), B5 (705 nm), B6 (740 nm), B7 (783 nm), B8a (865 nm), B11 (1610 nm) and B12 (2190 nm), with level L2A processing and surface reflectance (atmospheric corrected).

(d)Characteristics used for training and validation

The characteristics extracted from the nine spectral bands for each of the training and validation samples were the mean values and two vegetation indices: the NDVI [12] and the WDVI [13].

NDVI has been used by several authors [29,30] to monitor vegetation, and it is calculated by the difference between the near infrared (B8A) and red (B4) bands divided by the sum between bands B8A and B4, as shown in Equation (1).
NDVI = (B8A − B4)/(B8A + B4)(1)

The WDVI reflects the vegetation cover that minimizes the effects of wet bare soil. The mathematical expression for WDVI is shown in Equation (2).
WDVI = B8A − *a**B4(2)
here, *a* is the slope of the soil line; *a* was calculated using Equation (3).
(3)$$a=\overset{n}{\underset{i=1}{\sum }}B8A_{i}/\overset{n}{\underset{i=1}{\sum }}B4_{i}$$
here *n* is the number of plots without crop (bare land).

The final formula used to calculate the index is shown in Equation (4).
WDVI = B8A − 1.62*B4(4)

(e)Database

Visits to the plots were planned to coincide with the pass of the satellite. As such, each visit was scheduled on the date of the closest satellite scene, without exceeding 10 days of difference between the visit to the plot and the pass of the satellite.

The construction of the database was based on the works of Leite et al. [31] and Siachalou et al. [32], who use Hidden Markov models (HMMs). This method consists of exploring the spectral information of a sequence of satellite images to identify crops, since each crop has a specific temporal profile and the current state of a crop depends on its previous states. Therefore, spectral information varies for each phenological stage throughout the agricultural cycle, and the sowing dates between plots are not homogeneous. An example of this can be seen in Figure 7.

The period of 63 days (Figure 7) corresponds to the acquisition of three Sentinel-2 satellite images. The HMMs method assumes that the previous spectral information of the images differs for each crop and that in the time elapsed between one image and another, the changes present in the reflectance of the bands will be homogeneous for the same crop.

This method also assumes that the spectral information of the plot was recorded on 6 June. When evaluating previous dates (7 May and 2 April), there will be a greater similarity for the same crop and less similarity for different crops.

As shown in Figure 7, the reflectance of a plot is different if it is compared with the reflectance of another crop on the same date. However, this pattern can also occur when the same crop is sown on different dates, as shown in Figure 8.

Another limitation of the proposed method is that it will not always be possible to have a series of continuous satellite images free of clouds.

If a continuous series of cloud-free images is not available, it is difficult to form a homogeneous time series. Due to the presence of clouds in some images during the agricultural cycle, forming time series every 5 or 10 days to cover the entire cycle was impossible. However, it was possible to form time series every 15 or 30 days.

Under these conditions and considering that it will not always be possible to form the same combinations in other years or in other areas, four combinations were chosen to determine the smallest number of images necessary to generate classification models. The combinations used are shown in Table 4. Combinations C1 and C3 use three scenes and 33 descriptors (11 for each image), and combinations C2 and C4 use only two scenes and 22 descriptors.

A graphic description only for combination C2 y C4, during the months May, June and July, is illustrated in Figure 9. The presence of cloud in the images avoid the form combination with that date but it is possible in other dates. The Figure 9 shows how in this period model C4 is trained with nine observations for the same crop and how using the combination C2 six observations were possible.

Table 5 shows that when analyzing a particular date (such as 6 July), each recognition model is trained with a different number of scenes and/or scenes with different dates.

Table 6 shows the general structure of the database for combination C1. In this example, each plot was analyzed for a maximum of four possible phenological stages, depending on the date of emergence, which is the first criterion to be part of the database. The determination of emergence was performed by WDVI analysis. Thus, if the WDVI of the plot in any of the previous scenes was less than 0.005, the entire row was eliminated from the database, since this indicates that the crop did not exist (i.e., it was not visible).

Values of WDVI greater than 0.005 (which indicate crop emergence) were obtained by calculating the WDVI in plots visited no more than 10 days after emergence. Some of the crops used to obtain the values are shown in Figure 10.

(f)Classification method and algorithm used

Classification was performed using the “Classification Learner” module included in MATLAB software. One of the algorithms used was the SVM, multiclass method one vs. one and all of the predictors standardized, the Kernel Function was polynomial of order 3.

The basic form of this algorithm is expressed by Equation (5).
(5)f(x)=sgn{w·x+b}
where ***w*** is a weight vector and *b* is a threshold.

For the linearly separable case, a separating hyperplane can be defined for the two classes as: ***w*** · ***x***_i_ + *b* ≥ +1 (for ***f(x_i_)*** = +1) and ***w*** · ***x_i_*** + *b* ≤ −1 (for ***f(x_i_)*** = −1).

To allow separation decisions in hyperplanes, the following equation was used [9].
(6)$$f\left(x\right)=sgn\;\left\{\overset{r}{\underset{i=1}{\sum }}\alpha_{i}y_{i}k\left(x,x_{i}\right)+b\right\}$$
where *α_i_* is a Lagrange multiplier, *r* is the number of samples, ***k***(***x***, ***x_i_***) is a kernel function, ***x*** and ***x_i_*** are vectors in the input space and *b* is a model parameter.

The type of kernel used in this research was the cubic polynomial, which is described by Equation (7), *d* represents the order of polynomial (*d* = 3) and *c* = 1.
***k***(***x***,***x_i_***) = [(***x^T^***,***x_i_***) + *c*]*^d^*(7)

The second algorithm used was the BT with 100 packed trees; the results of this algorithm were evaluated with five-fold cross-validation [11].

(g)Validation of the models on same crop cycle

The models trained with the field data were validated with data from 140 plots (77 bean plots, 36 alfalfa plots and 27 bean plots) in the 2019 agricultural cycle, and all of the phenological stages within the agricultural cycle were classified. Once the classification was obtained, it was evaluated by recording the results in a confusion matrix to obtain the overall accuracy (OA) which is the total number of correctly classified cluster divided by the total number of clusters; producer’s accuracy (PA) is the fraction of correctly classified clusters with regard to all cluster of that ground truth class; and user’s accuracy (UA) is the fraction of correctly classified cluster with regard to all clusters classified as this class.

To determine the level of agreement between the true samples and the results of the classification model (excluding agreement exclusively attributable to random), the kappa coefficient was calculated [27].

The values of the kappa coefficient can vary from −1 to 1; maximum possible agreement corresponds to a kappa coefficient of 1; a kappa coefficient of 0 is obtained when the observed agreement is precisely what is expected due exclusively to random [33]; and a negative value indicates a negative association. Altman [34] performed a qualitative classification of the kappa coefficient according to its value, qualifying as poor if the kappa value was <0.2; regular if the kappa value was between 0.21 and 0.4; moderate if the kappa value was between 0.41 and 0.60); good if the kappa value was between 0.61 and 0.80; and particularly good if the kappa value was between 0.81 and 1.

(h)Testing of the models in the next agricultural cycle

The models, trained with data from the 2019 crop cycle, were tested in the 2020 crop cycle using the models and following the same restrictions described for the models evaluated. Figure 11 show the number and dates of the images used.

To assess the accuracy of model for recognize corn, alfalfa, and bean crops during the 2020 cycle, biweekly reports of irrigated and sown plots were used and recorded in vector layer format.

For the validation of the models, the whole polygons of the vector layer of plots were used to extract the features from the images.

At total of 895 plots were used for test the models (264 plots of alfalfa, 41 plots of bean, and 590 plots of corn).

## 3. Results and Discussion

This work validated the model on two data sets, the first on the validation samples of the same crop cycle (2019) discussed in Section 3.1 and Section 3.2 and the second validation was on the plot polygons of the irrigation module (SIG) in the next crop cycle

### 3.1. Results Obtained with the SVM Algorithm

Table 7 shows the results of the performance of the models using SVM for the different database structures. Combinations C1 and C3 (which included databases with three scenes) had similar overall accuracies (94.8% and 94.4%, respectively), and the highest kappa coefficient (0.91) was observed for both combinations. In contrast, the C4 combination resulted in the lowest precision (91.5) and a kappa coefficient of 0.86 since only two scenes and a smaller timeframe were used. The difference between the highest and lowest precision was 3.3%.

In general, all of the crop recognition models had a kappa coefficient greater than 0.85; according to Altman [34] and Alagic et al. [33], the models received a rating of particularly good. Therefore, for all of the combinations, there was particularly good agreement between the reference data and the results of the model, which is supported by the observed accuracies of more than 90%.

When analyzing the accuracies by crop, maize was the best classified in the four models; the lowest classification precision was obtained for bean cultivation. This is similar to what was found by Kussul et al. [35], who analyzed eight crops, including corn and soybean; the soybean plant, which is similar to the bean plant, had one of the lowest precisions of all of the crops. The UA of maize and beans were lower than the PA since the models confused the first phenological stages of maize and beans since these crops have similar spectral reflectance [36]. In the case of alfalfa, the UA was higher than the PA since alfalfa has a particular developmental pattern which is easily distinguished from those of corn and beans.

### 3.2. Results Obtained with the BT Algorithm

With the BT algorithm, the model using the C1 combination obtained the highest OA and kappa coefficient, and the lowest OA and kappa coefficient were observed for combination C4 (Table 8). The difference between the highest and lowest precision values was 6.9%. According to the classification described by Alagic et al. [33], the results of the classification model with combinations C1 and C3 were particularly good, indicating agreement between the true data obtained in the field and the data resulting from the recognition of the crops in satellite images. The kappa coefficients were greater than 0.80 and the OA values were greater than 90%. The combinations C2 and C4 had good consistency since their kappa coefficients were greater than 0.61 and their OA values were greater than 84.9%. It is important to note that in combinations C1 and C3 three scenes of satellite images were used. In combinations C2 and C4, only two scenes were used. The combination with three images had better OA.

When we analyzed the precision of each crop, we observed that the bean crop had the lowest overall precision according to both the UA and PA. The highest accuracy of the classifiers was 81.6, and the lowest was 74.5.

### 3.3. Comparison of the Results Obtained with the Two Algorithms: SVM and BT

Table 9 compares the results obtained by the two classification algorithms used. Among the four combinations, the SVM model resulted in the highest OA and kappa coefficient values. These results agree with those obtained by Rahman et al. [37], who tested three algorithms, including bagged trees and support vector machine, for the classification of soils and found that SVM was the classification model with the best accuracy. Chakhar et al. [38] evaluated multiple algorithms for crop classification, including the two models used in this work, and found that SVM performed slightly better. However, other authors, such as Adam et al. [39], found that land cover and land use classifications made with random forest (RF) were slightly better than those made with SVM. Similarly, Löw et al. [40], who classified crops by combining both methods, found that RF performed slightly better, instead Mardani et al. [41] rated BT as slightly superior to other models.

The reason that the SVM method produced slightly better results may be due to database features fit better to the user-defined parameters during training [42], but in general, good results were obtained from both methods.

As shown in Table 9, the differences in OA between the methods ranged from 2.9% for C1 to 6.5% for C4. The classification performance improved when using a greater number of scenes or a larger timeframe. On average, the difference in OA was 5% considering the four combinations.

### 3.4. Test of the SVM Model in the Subsequent Cycle

After evaluating the two models (BT and SVM) and finding that the SVM model had the best overall accuracy and the best kappa coefficient, the SVM approach was chosen to carry out the test of the models of the four combinations of databases on the next crop cycle. The models were tested on the 2020 crop cycle, and the results are shown in Table 10, Table 11, Table 12 and Table 13.

Note that each table has different number of combinations and each combination use a different number of scenes or different scene dates. Also, it should be noted that the dates selected for validation depended on cloud-free images.

The classification carried out for 6 May (Table 10) had moderate agreement, with a kappa coefficient of 0.53 [33]. For all of the subsequent dates, there was good agreement, with kappa coefficients ranging between 0.63 and 0.74 [33]. The low kappa coefficient observed was since a satellite image acquired 21 April 2020 was used as the first scene, when most of the plots, including the corn and bean plots, were in the initial stages of growth. The same result was found for combinations C2 and C3, in the classification carried on 21 May (Table 11).

When analyzing model precision by crop, it was concluded that the SVM model was not capable of recognizing bean crops. The accuracy of the classifier for this crop was in the range of 24.3 to 41.2%, indicating that it was not reliable for classifying beans. The reason the model did not learn to classify beans is that the model was trained (2019 crop cycle) in an area in which bean planting was limited. Therefore, the number of samples (plots) was not sufficient, the number of plots is very unbalanced, with only 41 plots for beans and 590 for corn. However, the SVM model was capable of recognizing corn and alfalfa in satellite images in other periods.

Table 10 shows the results for combination C4, which was the only combination of the available images; this combination had the lowest OA and kappa coefficient values. At the beginning of the agricultural cycle, there was relatively high confusion among the three crops. Maize was confused with beans since the plots had larger areas of bare soil.

The results in Table 11 show an improvement in both the OA and the kappa coefficient. If we compare combination C4 for 6 May with that for 21 May, the OA and kappa coefficient improved by 8.1% and 0.1, respectively. Combination C2 (with scenes grouped in 30-day periods) had the best overall accuracy and kappa coefficient. The best classified crop in all of the combinations for this date was corn, with a UA of 95.5%.

Table 12 shows that the OA and kappa coefficient values improved by an average of 4% and 0.06, respectively. It is important to note that despite the problems associated with adequately classifying the bean crop, such as the small cultivated area (and therefore, few samples for training), the global accuracy of the classifier for the three models was greater than 85%. Therefore, these models were classified as efficient [43].

Table 13 shows the four models used to classify the alfalfa and bean crops on 20 June 2020. On this date, the model that performed the best used combination C2, with a kappa coefficient of 0.74 and OA of 87.5%.

As the agricultural cycle progressed, the precision of the classification improved (Figure 12). When using any of the combinations evaluated, acceptable classifications (greater than 79%) of corn and alfalfa were observed at the beginning of June, providing decisionmakers with information related to the initial stages of cultivation.

Comparing the combinations on each of the dates (Figure 12) revealed that despite not having many scenes, the accuracies of the models did not vary more than 3% between combinations on the same date. Therefore, if combination C4 used only two images (the current image and an image from 15 days (about 2 weeks) prior), this was the most highly recommended combination from the beginning of June, with a global accuracy greater than 85%. In contrast, Leite et al. [31] found that the accuracy in the classification improved with an increasing number of images and showed that with a sequence of 3 images, the accuracies were greater than 80%.

Even though the precision of the bean classifications did not exceed 41.2%, the general classification was not affected since the number of parcels with beans grown in the irrigation module did not exceed 5% of the total module (with the most important crops being corn and alfalfa). This is also reflected in the kappa coefficient; except for 6 May, the kappa coefficient values were greater than 0.63, reaching 0.74 in some cases, which indicates good consistency according to Cohen [44].

The results of this work suggest that a larger number of images increases accuracy, which coincides with the results presented by Leite et al. [31]. Therefore, if images acquired by the Sentinel-2 sensor were available every five days, the results of the model would be better. However, obtaining images at this frequency for the entire agricultural cycle is complicated by the presence of cloud cover.

The classification accuracies obtained are similar to those reported by Zheng et al.; Hegarty-Craver et al., Saini and Ghosh [15,17,22] and slightly smaller than those found by Yang et al. [14] and Tran et al. [25]. The methodology shows similarities with the classifications that Martinez et al. [16] and Blickensdörfer et al. [26] used with time series, with the difference being that these generally use more than three images. The difference with most of them is that the result is only a map of crops valid for the entire agricultural cycle, so it is necessary to carried out field work again to replicate the methodology.

The use of time series of optical images has the difficulty for replicating or saving the model, due to presence of clouds, so a continuous series cannot be formed, even Tran et al. [25] use gap filling to counteract this problem. In this research, the implemented methodology addresses this problem by generating models for a combination of available images and using only two or three images. Leite et al. [31] found similar results with two images unlike with time series; Conrad et al. [21] achieves overall accuracies of 80% with two Aster images, the difference with Leite et al. [31] is that the training samples were selected by a well knowing expert and Conrad makes field visits for only two months in the agricultural cycle.

The most similar works that try to show classifications before the end of the agricultural cycle without having to go to the field are Cai et al. [20], Konduri et al. [24], Lin et al. [19] and Defourny et al. [18]. Some use statistics from the past few years to find patterns and determine the established crop. However, this methodology cannot be used in all countries, as is the case in Mexico, since there is not as much historical georeferenced information of crops. On the other hand, Defourny et al. [18] exposes some results the project Sen2-Agri system allowing for national scale automated agricultural monitoring.

With respect to near real time classification, our results are similar to that of Lin et al. [19] and Kounduri et al. [24]. Lin et al. [19] obtains an accuracy greater than 80 when the corn is in the maturation stage and the soybean begins to flower and Kounduri et al. [24] has precision greater 70% for corn and soybean, these precisions increase as the agricultural cycle is completed. Finally Cai et al. [20] reaches a precision of >95% for beans and soybeans.

In the present work, the precisions for the crops increase in June when the crops are in the maturity stage. It would possibly be better in July, but in July most of the beans have been harvested, and our model considers that the crop must be standing and not harvested.

There are two novelties in the proposed approach, the first is the models that use small discontinuous series of two or three images, these combinations of images can be searched at any date within the agricultural cycle and the second is that the models collect temporal patterns throughout the crop cycle and not just one for the entire cycle.

The limitations for replicating this work is that for training, the models are necessary for field surveys before it is necessary to identify the plot with bare soil and thus calculate the slope of soil (which is necessary for the WDVI index). The WDVI index is helpful since it indicates the presence of vegetation. The same indicator for crop presence is used since the samples are cultivated plots. However, if the classification is carried when the crop had been harvested the model will not consider this plot since the model was only trained with standing crops. Another limitation is that an updated vector layer of plots is required for assembling the database for input in the models; the number of combination possible will depend of the number of cloud-free images, and if there are an excess of images with clouds, not all patterns can be evaluated during the cycle.

Future directions of research to improve this classification method include increasing the number of classes for the same crop (i.e., bean in growth phase and bean in maturity phase) and to include other classes such as bare land, forest, and urban settlements with the aim of generating thematic maps.

## 4. Conclusions

The machine learning algorithms SVM and BT yielded classifications of corn, alfalfa, and bean crops with good accuracy (mainly for corn and alfalfa).

The model generated for the study area reach the best accuracy in the third month of the agricultural cycle using two images and a timeframe of 30 days (combination C2).

The best time for classification within the cycle is in the middle of the agricultural cycle, when crops are in maturity stage with only two images and there is a time window between the images taken at 30 days.

As the sequence of images covers all of the phenological stages of the crops, the models can be used at any date within the agricultural cycle, assuming that the necessary cloud free images are available to achieve the required image combinations.

## Figures and Tables

**Figure 1 sensors-22-06106-f001:**
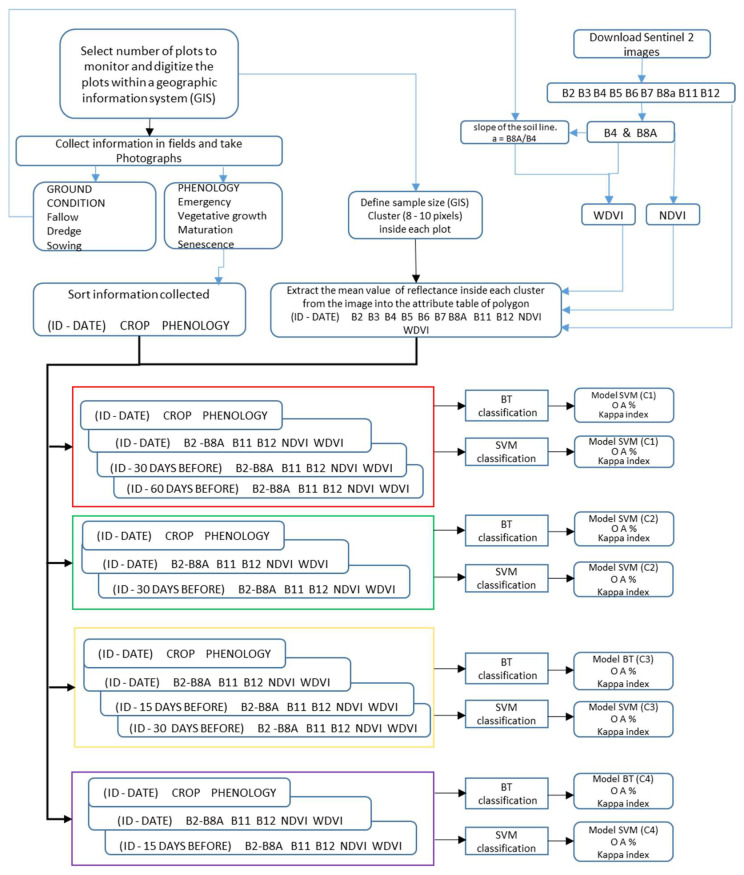
General overview of the methodology used to obtain the models for crop mapping.

**Figure 2 sensors-22-06106-f002:**
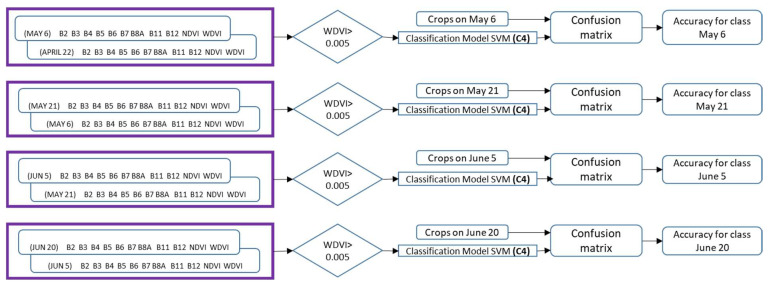
Steps followed in the validation on the next crop cycle using the SVM classification model on different dates only with combination 4 (C4).

**Figure 3 sensors-22-06106-f003:**
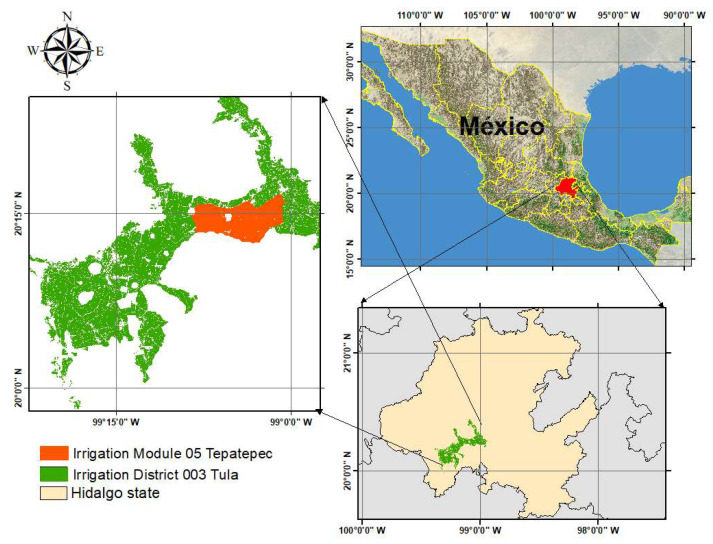
Geographic location of the study area.

**Figure 4 sensors-22-06106-f004:**
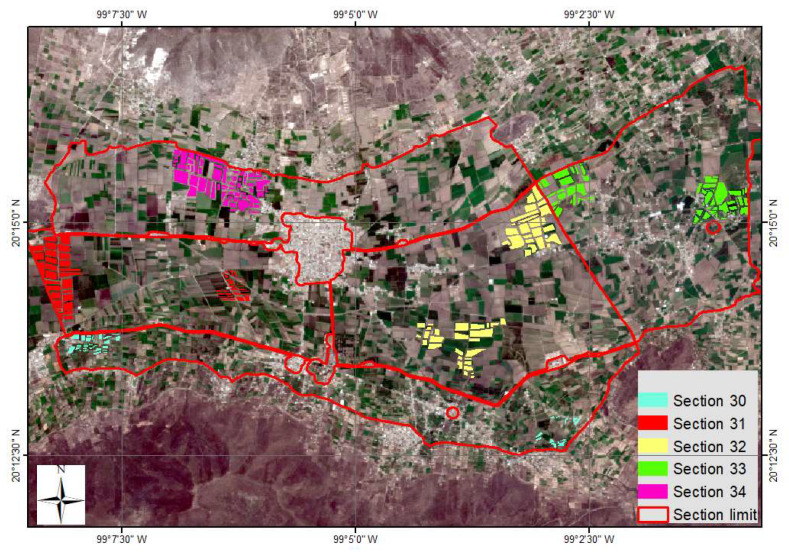
Parcels (clusters) by section grouped and monitored during the crop cycle (2019).

**Figure 5 sensors-22-06106-f005:**
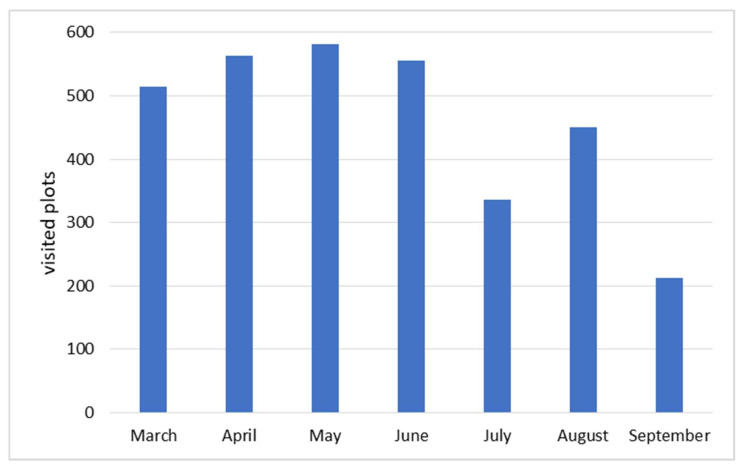
Number of parcels visited per month during the 2019 crop cycle.

**Figure 6 sensors-22-06106-f006:**
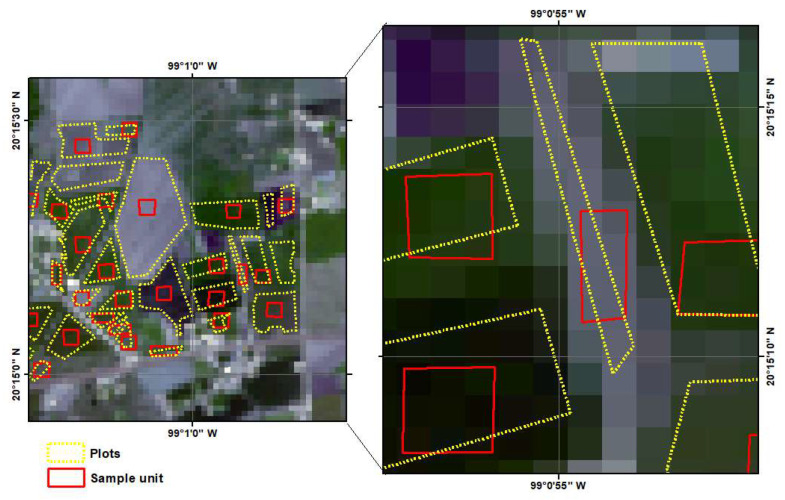
Sample sizes: 3 × 3 pixels (**left**), 2 × 5 pixels (**center**) and 2 × 4 pixels (**right**).

**Figure 7 sensors-22-06106-f007:**
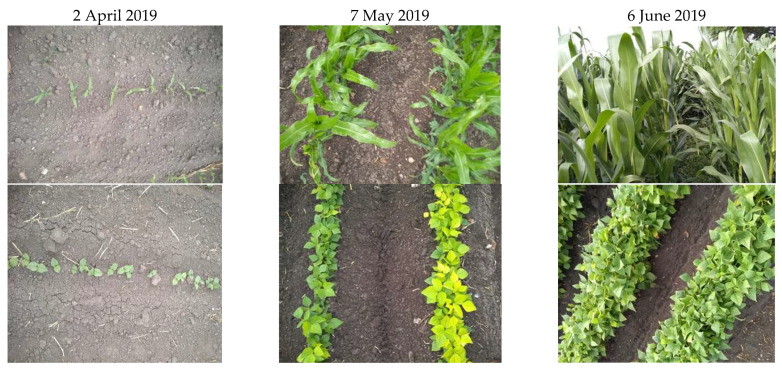
Temporal profiles of corn (**top**), beans (**center**) and alfalfa (**bottom**) over a period of 63 days (about 2 months).

**Figure 8 sensors-22-06106-f008:**
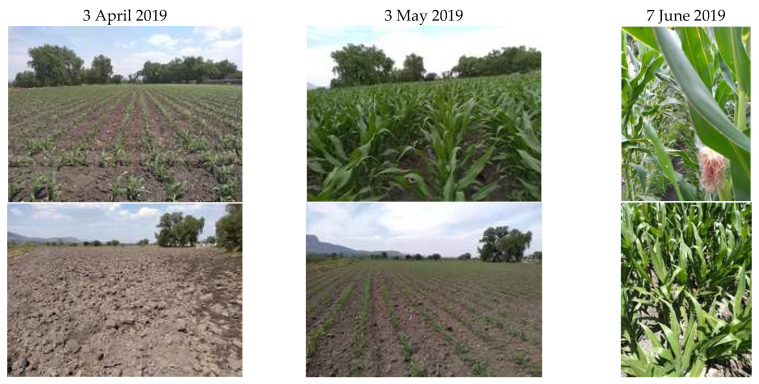
Differences in the sowing date of the same crop (corn).

**Figure 9 sensors-22-06106-f009:**
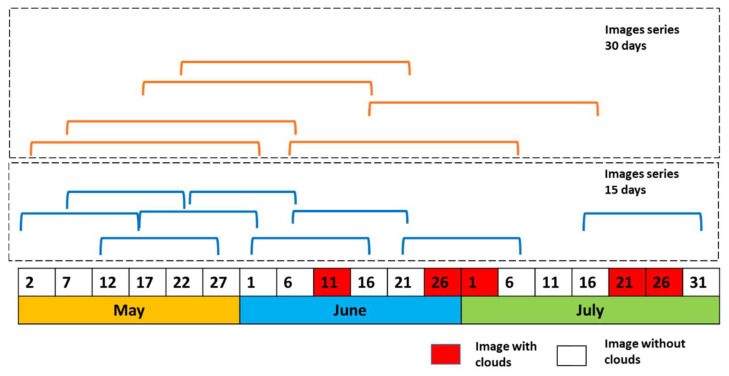
Images used for combination C2 and C4 for May, June, and July.

**Figure 10 sensors-22-06106-f010:**
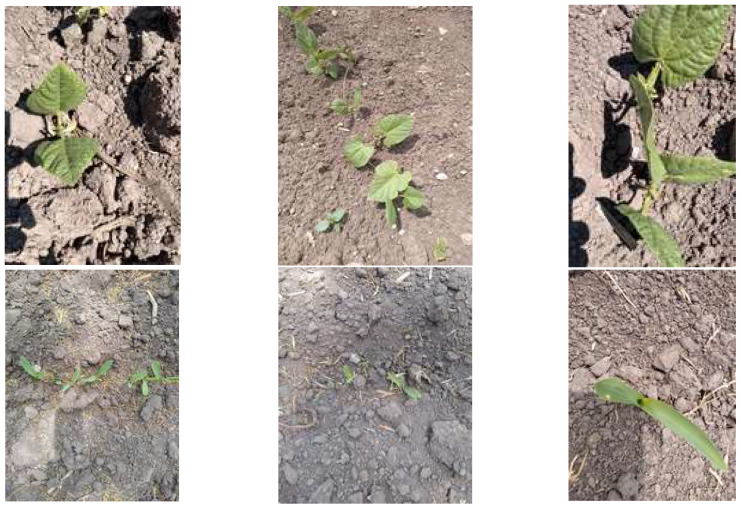
The threshold WDVI value for crop presence was determined based on the established state of the plots during regular visits no later than 10 days after emergence.

**Figure 11 sensors-22-06106-f011:**
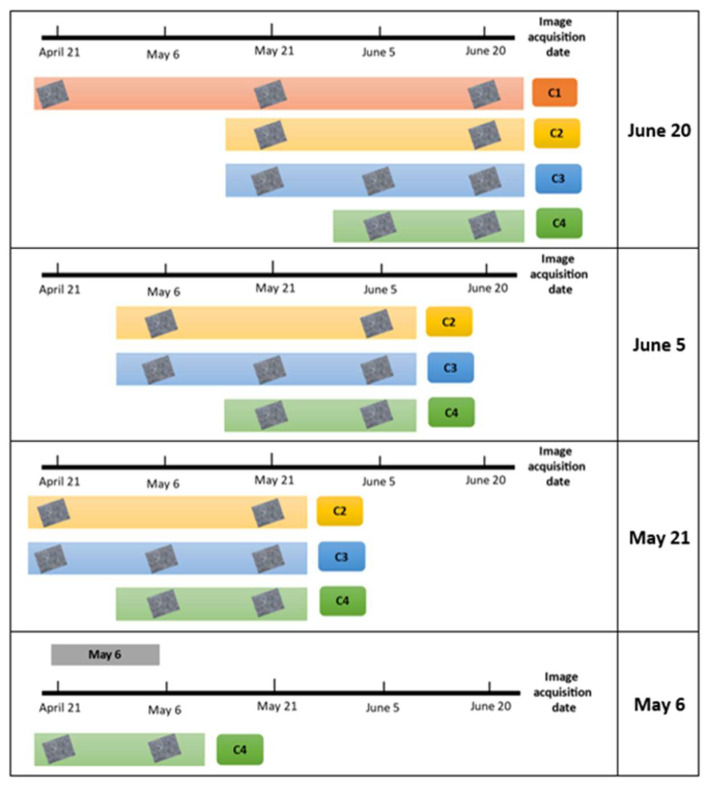
Number and date of images used to validate the models in the next crop cycle.

**Figure 12 sensors-22-06106-f012:**
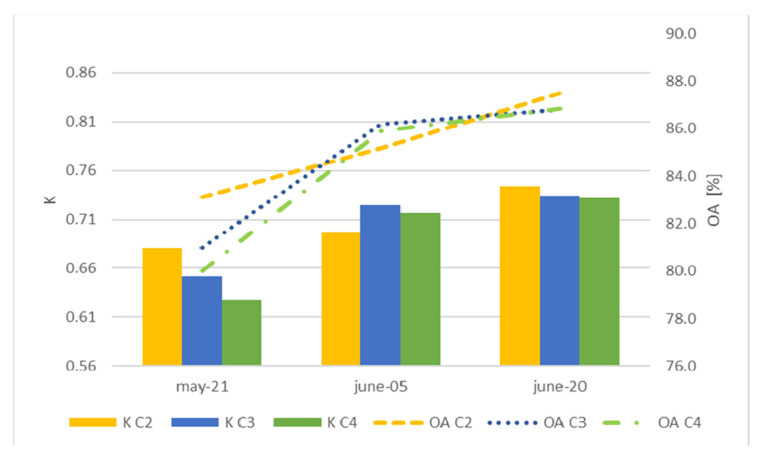
Overall accuracy (OA) and kappa coefficient (K) for each combination used. C2: 2 scenes every 30 days (about 4 and a half weeks). C3: 3 scenes every 15 days. C4: 2 scenes every 15 days.

**Table 1 sensors-22-06106-t001:** Some methodologies and techniques applied to crop mapping.

Author	Range of Images	Images	Training Samples	Classification Method	Within-Season Mapping
Martínez et al. [16]	multi-year	Landsat TM y ETM	Fields survey	Cross classification (IDRISI sotware)	No
Zheng et al. [17]	Agricultural year	Landsat TM y ETM	Fields survey	SVM	No
Conrad et al. [21]	Crop Cycle	SPOT 5 y ASTER	Expert knowledge	Rules of classification	No
Saini and Ghosh [15]	Crop Cycle	Sentinel 2	Fields Survey	RF and SVM	No
Yang et al. [14]	Crop Cycle	SPOT 5	Fields survey	MD, M-distance, MLE, SAM and SVM	No
Hegarty-Craver et al. [22]	Crop Cycle	Image UAV, Sentinel 1 y Sentinel 2	Fields survey	RF	No
Prins and Van Niekerk [23]	Crop Cycle	Aerial Image, LIDAR image, Sentinel 2	Crop type database	RF, DTs, XGBoost, k-NN, LR, NB, NN, d-NN, SVM-L, and SVM RBF	No
Cai et al. [20]	Multi-year	Landsat TM y ETM	CDL (USDA)	d-NN	Yes
Konduri et al. [24]	Multi-year	MODIS	CDL (USDA)	unsupervised classification (phenoregions)	Yes
Lin et al. [19]	Multi-year	Sentinel 2, Landsat 8	CDL (USDA)	RF	Yes
Tran et al. [25]	Crop Cycle	Sentinel 2	CDL (USDA)	RF	No
Blickensdörfer et al. [26]	Multi-year	Sentinel 1 Sentinel 2 Landsat 8	Land Parcel Information System (LPIS).	RF	No
Defourny et al. [18]	Multi-year	Sentinel 2	Ground truth data	RF	Yes

Support vector machine (SVM); random forest (RF); minimum distance (MD); Mahalanobis (M-distance); maximum likelihood (MLE); spectral angle mapper (SAM); decision trees (DTs); extreme gradient boosting (XGBoost); K-nearest neighbor (k-NN); logistic regression (LR); naïve Bayes (NB); neural network (NN); deep neural network (d-NN); SVM with a linear kernel (SVM-L); SVM with a radial basis function kernel (SVM RBF); Crop Data Layer (CDL).

**Table 2 sensors-22-06106-t002:** Crop patterns of the main species planted in the Tepatepec Irrigation Module.

Crop	March				April				May				June				July				August				September				October			
Bean																																






Corn																																






Alfalfa																																






**Table 3 sensors-22-06106-t003:** Acquisition dates of the scenes included in the database.

Month	Available Images	Images Used	Image Acquisition Date						
April	6	6	2	7	12	17	22	27	
May	6	6	2	7	12	17	22	27	
June	6	4	1	6	11	16	21	26	
July	7	4	1	6	11	16	21	26	31
August	6	6		5	10	15	20	25	30
September	6	1		4					

**Table 4 sensors-22-06106-t004:** Combinations and number of scenes used in each combination.

Combination	Number of Descriptors	Number of Images	Dates of Scenes Used		
C1	33	3	CD	30 DB (6 PS)	60 DB (12 PS)
C2	22	2	CD	30 DB (6 PS)	
C3	33	3	CD	15 DB (3 PS)	30 DB (6 PS)
C4	22	2	CD	15 DB (3 PS)	

CD, current date; DB, days before; PS, previous scenes.

**Table 5 sensors-22-06106-t005:** Example of the dates and number of scenes used for each possible combination.

Combination	Scenes Used		
C1	6 July 2019	6 June 2019	7 May 2019
C2	6 July 2019	6 June 2019	
C3	6 July 2019	21 June 2019	6 June 2019
C4	6 July 2019	21 June 2019	

**Table 6 sensors-22-06106-t006:** Example of the database structure for combination C1.

Cultivation Type	Date of Analysis	Dates of Images or Scenes Included in the Database		
corn 1	6 September	4 September	5 August	6 July
corn 1	5 August	5 August	6 July	6 June
corn 1	6 July	6 July	6 June	7 May
corn 1	6 June	6 June	7 May	7 April
corn 77	4 September	4 September	5 August	6 July
corn 77	5 August	5 August	6 July	6 June
corn 77	6 July	6 July	6 June	7 May
corn 77	6 June	6 June	7 May	7 April

**Table 7 sensors-22-06106-t007:** Accuracy assessment obtained in each of the crop recognition models with the SVM algorithm.

Combination	Kappa Coefficient	Overall Accuracy %	Corn		Alfalfa		Bean	
			PA %	UA %	PA %	UA %	PA %	UA %
C1	0.91	94.8	97.2	96.0	94.1	97.4	89.2	87.5
C2	0.89	93.4	95.8	95.1	94.6	96.0	85.2	84.9
C3	0.91	94.4	96.6	95.0	93.9	98.0	89.0	87.5
C4	0.86	91.5	94.3	93.3	90.5	97.4	85.1	80.4

**Table 8 sensors-22-06106-t008:** Accuracy assessment obtained for each of the crop recognition models with the BT algorithm.

Combination	Kappa Coefficient	Overall Accuracy %	Corn		Alfalfa		Bean	
			PA %	UA %	PA %	UA %	PA %	UA %
C1	0.87	91.8	95.8	95.9	89.3	91.8	84.9	81.3
C2	0.79	87.4	94.2	90.4	87.9	88.3	67.6	75.9
C3	0.84	90.6	95.1	91.5	93.1	94.1	73.8	81.6
C4	0.74	84.9	93.0	85.2	89.6	90.9	57.8	74.5

**Table 9 sensors-22-06106-t009:** Accuracy assessment of the algorithms used as a function of the acquisition dates of the scenes used.

3 scenes			C3		C1	

			SVM	BT	SVM	BT
	Kappa coefficient		0.91	0.84	0.91	0.87
	Global accuracy		94.4%	90.6%	94.8%	91.8%
2 scenes	C4		C2			

	SVM	BT	SVM	BT		
	0.86	0.74	0.89	0.79	Kappa coefficient	
	91.5%	84.9%	93.4%	87.4%	Overall accuracy	
	15 days		30 days		60 days	

**Table 10 sensors-22-06106-t010:** Results of the classification on 6 May in the 2020 agricultural cycle.

Combination	Kappa Coefficient	Overall Accuracy %	Corn		Alfalfa		Bean	
			PA %	UA %	PA %	UA %	PA %	UA %
C4	0.53	71.9	67.2	94.3	80.5	82.4	77.1	14.2

**Table 11 sensors-22-06106-t011:** Results of the classification on 21 May in the 2020 agricultural cycle.

Combination	Kappa Coefficient	Overall Accuracy %	Corn		Alfalfa		Bean	
			PA %	UA %	PA %	UA %	PA %	UA %
C2	0.68	83.1	81.7	95.5	89.3	76.2	57.1	30.8
C3	0.65	81.0	78.5	95.3	85.5	80.6	80.0	25.2
C4	0.63	80.0	77.7	95.0	83.3	80.3	82.5	24.3

**Table 12 sensors-22-06106-t012:** Results of the classification on 5 June in the 2020 agricultural cycle.

Combination	Kappa Coefficient	Overall Accuracy %	Corn		Alfalfa		Bean	
			PA %	UA %	PA %	UA %	PA %	UA %
C2	0.70	85.2	89.2	91.2	81.1	84.6	52.6	32.3
C3	0.72	86.2	86.8	95.1	88.6	79.0	60.5	40.4
C4	0.72	85.9	87.5	93.9	86.7	80.6	56.4	37.3

**Table 13 sensors-22-06106-t013:** Results of the classification on 20 June in the 2020 agricultural cycle.

Combination	Kappa Coefficient	Overall Accuracy %	Corn		Alfalfa		Bean	
			PA %	UA %	PA %	UA %	PA %	UA %
C1	0.70	84.4	82.6	95.5	90.5	79.5	62.1	28.6
C2	0.74	87.5	89.4	94.0	88.2	83.5	51.4	36.7
C3	0.73	86.8	86.9	94.9	89.7	80.0	61.8	41.2
C4	0.73	86.9	88.2	94.0	87.5	83.4	58.8	35.1

## Data Availability

The data presented in this study are available on request from the corresponding author.

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
