# Peer review of "Monitoring and Identification of Agricultural Crops through Multitemporal Analysis of Optical Images and Machine Learning Algorithms"

_sensors, 2022, doi:10.3390/s22166106_

Round 1

Reviewer 1 Report

In this paper, the authors have studied the monitoring and identification of agricultural crops through machine learning algorithms. Overall, the article is well written but requires some major improvements.

1) Only 1 dataset is studied. consider including at least 1 more dataset in your study.

2) Separate related work from the introduction section. In related work include the comparison table that should highlight the strengths and weaknesses of the proposed method as well as previous methods.

3) Please write the complete manuscript in English instead of your national language specifically check Table 1 & 3, Figures 3 & 9, etc.

4) What was the motivation behind the selection of SVM and BT classifiers? Why other classifiers are totally neglected? 

5) After the introduction section, include your contributions to this manuscript in 3-4 bullets. 

Author Response

Dear Reviewer

First of all, I would like to thank you for your insightful comments, which will undoubtedly greatly improve the manuscript.

The following is an explanation of how they were treated.

Comments and Suggestions for Authors

Responses to the referees comments.

Review1

1) Only 1 dataset is studied. consider including at least 1 more dataset in your study.

In the first tests of the model we combined the data set removing some characteristics such as the number of bands or the indices, the variation was smaller but not on a large scale and always with less precision in the end. What we want to highlight in this paper is the proposed methodology which take sub data set that can be formed throughout the agricultural cycle. Figure 1 and Figure 2 have been added to the paper, which describe the methodology general to obtain the models and their test and validation during the following agricultural cycle.

2) Separate related work from the introduction section. In related work include the comparison table that should highlight the strengths and weaknesses of the proposed method as well as previous methods.

We attended your proposal and added a section in the paper, some older works compared to the most recent ones on crop mapping were put.

3) Please write the complete manuscript in English instead of your national language specifically check Table 1 & 3, Figures 3 & 9, etc.

The figures and tables were corrected and written in English

4) What was the motivation behind the selection of SVM and BT classifiers? Why other classifiers are totally neglected?

BT, which is very similar to Random Forest and SVM, are the two algorithms that are mentioned the most when analyzing data with machine learning and they are the ones that achieve the best results in terms of classification accuracy when compared to others. Another algorithm also used and with good results in terms of accuracy is artificial neural networks, but they have the characteristic of working better with a large amount of data, something that we did not have in this work.

5) After the introduction section, include your contributions to this manuscript in 3-4 bullets.

We attended your proposal and mentioned the contributions of this work and its disadvantage in a section

Does the introduction provide sufficient background and include all relevant references?

New references were added that help support the work

Is the research design appropriate?

A figure was added where shown the general methodology followed to obtain the classification models and another one wich shows how this are validated in the next crop cycle, this in order to help clarify the steps followed in the work

Reviewer 2 Report

Review of paper "Monitoring and identification of agricultural crops through multitemporal analysis of optical images and machine learning algorithms"

The paper describes the process of crop classification from Sentinel-2 satellite images. The procedure is technically sound, and the text is quite readable. However, there are several major issues with the current presentation.

The main issue with the paper is that it does not explain how the presented research improves or even relates to existing crop classification models. The paper does not contain any major novelties and original contributions, but seems to apply established methodologies to different data. There is basically no related work section, only a few cited papers in the results section. There is no comparison of results with previous work. Crop classification using Sentinel-2 imagery is a well researched problem, and there are many missing references, to give only a few:
- Sonobe et al., Crop classification from Sentinel-2-derived vegetation indices using ensemble learning, 2018
- Saini & Ghosh, Crop classification on single date sentinel-2 imagery using random forest and support vector machine, 2018
- Yi et al., Crop classification using multi-temporal Sentinel-2 data in the Shiyang River Basin of China, 2020
- Seydi et al., A Dual Attention Convolutional Neural Network for Crop Classification Using Time-Series Sentinel-2 Imagery, 2022
- Pluto-Kossakowska, Review on Multitemporal Classification Methods of Satellite Images for Crop and Arable Land Recognition, 2021

The next major problem I had was understanding the overall structure, e.g., how the training samples are generated, or how the classification is actually performed. The method description is scattered across multiple sections, but one does not get the high-level overview of model inputs and outputs. Below are specific major issues the authors should address if paper revision is considered:

Section b is crucial for understanding the method, but there is a lot of missing information:
- On page 5, it is stated that the sampling unit is a group of pixels, which can be of variable shape. How do you determine these pixel groups, and why do you need them in the first place? Why not take the whole parcel as a sample and classification unit, since that information is obviously available?
- How are the input features for the classifier calculated from band values - in line 174, it is stated that "statistics corresponding to the reflectance are determined". Which statistics?

Section d on page 6:
- Page 6, line 205: what is meant by "the ordinary moments of order"? Which characteristics are extracted?
- How are the features from different time points combined - are they simply concatenated into a training sample?
- What is the final size of the feature vector for the classifier?

Section e:
- on page 8, line 264 - what does it mean that the satellite images were "grouped" by 30-day or 15-day periods? If there are multiple images in a group, and the image on the exact date is not clear, is a replacement image from the group used?
- Table 6 - what is the role of the "date of analysis" column, does it always coincide with the last image date?

Section g, page 11:
- line 334: What does it mean that "all the phenological stages within the agricultural cycle were classified" - were separate classifiers trained for each phase, or was there a single classifier trained on all of the data?
- line 337: the term "global precision" is wrong, the authors are evaluating overall accuracy (OA); later in line 376 they call it "global accuracy", only in table 9 the term "overall accuracy" is used
- what is "precision of reference samples"? why is the acronym PMR?
- "precision of classifier" should be explained as well; also, precision metric is normally accompanied by recall and F1, because it can be trivially boosted in isolation; if PC really refers to classifier precision, recall and F1 should also be reported
- SVM is inherently a binary classifier, it should be noted what method of multi-class methodology was used here (e.g., one-vs-rest)?

Section h, page 11, lines 365-366: "To enhance the database, the values of the pixels in the satellite images that made up each plot were averaged." - why is this process different from the one for preparing the training data? Why does that "enhance the database"?

It should be explicitly emphasized that the results in sections 3.1-3.3 are validation results for the samples from the same season as the training data.

Page 12, section 3.1:
- precision and accuracy are mixed in the text, the authors should check the definitions of accuracy and precision metrics for binary classifiers, and carefully correct all of the references to proper terms
- in Table 9, text at the bottom, is another source of ambiguity: PC is explained as classifier accuracy, before it was defined as precision of classifier

Page 13, section 3.2:
- in line 404, what is PG? before it was GP, which is itself wrong, as explained before
- in lines 409-413, the "while" part of the sentence does not follow from the first part of the sentence, since it is talking about a different aspect

Page 13, section 3.3:
- Figure 8 should in fact be Table 11, and should not be split across pages; the table structure is also messed up
- the statement in lines 435-436 is speculative and not convincing - what does the DB structure have to do with classification efficiency?

Page 14, section 3.2:
- the section number is wrong, it should be 3.4
- the contents of tables 11-14 are unclear - what does it mean that the "combination includes image acquired on May 6" and "classification performed for the scene acquired on May 6"? As I understand, it is not the image itself that is being analyzed or classified, it is simply the last image in the input temporal sequence.
- Table 12 is split across pages and hard to inspect
- in line 490, accuracy is again used to refer to bean classifier "precision"

Bibliography should be thoroughly checked, e.g. in bibliography item [25] by Chakhar et al. the title is incomplete.

Minor issues:

Table 1 is in Spanish, its caption is on a previous page.

Page 2, line 81: the authors state that "the training samples result from the correlation between the field observations and the plots identified in the Sentinel-2 images" - what correlation are they talking about, and how does it define the training samples? I didn't find any mention of these correlations later on. What is the reason for periodically revisiting the plots besides determining the ground truth value for the classifier? These points is necessary to explain.

Are parcels in Figure 2 color coded in some way? This information should be added to the figure caption.

Figure 3 and Table 3 are in Spanish.

By substituting equation Eq. (3) into Eq. (2), we get WDVI=0, which is puzzling. I suggest to reformat the text and explain in advance that the value of a is calculated on bare land, then used on data during the crop season.

The information in lines 298-300 on page 9 is a bit unclear without reading the caption of Figure 7. I suggest to rephrase that the threshold WDVI value for crop presence was determined based on the established state of the plots during regular visits no later than 10 days after emergence.

Page 10, equation (5) and (6) - I suggest using bold for vectors and italic for scalars. Making notation consistent is also required. Explanations of r and x_i in Equation (6) are missing.

Page 10, Equation (7): How does the equation represent a cubic polynomial? What is "(x,x_i)", and what is the value of d? What is the interpretation of addition of 1 to a pair of vectors?

Author Response

RESPONSE TO REVIEWER 2

Dear Reviewer

First of all, I would like to thank you for your insightful comments, which will undoubtedly greatly improve the manuscript.

COMMENTS
The main issue with the paper is that it does not explain how the presented research improves or even relates to existing crop classification models. The paper does not contain any major novelties and original contributions, but seems to apply established methodologies to different data. There is basically no related work section, only a few cited papers in the results section. There is no comparison of results with previous work. Crop classification using Sentinel-2 imagery is a well researched problem, and there are many missing references, to give only a few:
- Sonobe et al., Crop classification from Sentinel-2-derived vegetation indices using ensemble learning, 2018
- Saini & Ghosh, Crop classification on single date sentinel-2 imagery using random forest and support vector machine, 2018
- Yi et al., Crop classification using multi-temporal Sentinel-2 data in the Shiyang River Basin of China, 2020
- Seydi et al., A Dual Attention Convolutional Neural Network for Crop Classification Using Time-Series Sentinel-2 Imagery, 2022
- Pluto-Kossakowska, Review on Multitemporal Classification Methods of Satellite Images for Crop and Arable Land Recognition, 2021

ANSWER
A section named “Related works was added for comparison and discussion”

The next major problem I had was understanding the overall structure, e.g., how the training samples are generated, or how the classification is actually performed. The method description is scattered across multiple sections, but one does not get the high-level overview of model inputs and outputs. Below are specific major issues the authors should address if paper revision is considered:

ANSWER
Figures were added
which describe the methodology general to obtain the models and their test and validation during the following agricultural cycle.

Section b is crucial for understanding the method, but there is a lot of missing information:
- On page 5, it is stated that the sampling unit is a group of pixels, which can be of variable shape. How do you determine these pixel groups, and why do you need them in the first place?

Why not take the whole parcel as a sample and classification unit, since that information is obviously available?

ANSWER

Yes we have the plot polygon but the size of plot are very different in the irrigation module so we decided to use same number of pixels for each plot and are important to draw the polygon because the attribute table of the shapefile will contain the values (average) of the reflectance from each band of the sentinel 2 image

- How are the input features for the classifier calculated from band values - in line 174, it is stated that "statistics corresponding to the reflectance are determined". Which statistics?

ANSWER

We use the zonal statics tool of QGIS that calculates some statistics values for pixels of input raster inside certain zones, the zones are defined by a polygon layer.

Section d on page 6:
- Page 6, line 205: what is meant by "the ordinary moments of order"? Which characteristics are extracted?

ANSWER

We extract the mean value (first order moment) of each band (reflectance), the redaction was changed
- How are the features from different time points combined - are they simply concatenated into a training sample?

Yes they are joined in a record of the database (table) that is the input of classifiers
- What is the final size of the feature vector for the classifier?
The final vector is different for each combination, C1 y C3 has the response variable and 33 descriptors, is the result of add the 11 descriptors of each image (B2, B3, B4, B5, B6, B7, B8a, B11, B12, NDVI Y WDVI)
Section e:
- on page 8, line 264 - what does it mean that the satellite images were "grouped" by 30-day or 15-day periods? If there are multiple images in a group, and the image on the exact date is not clear, is a replacement image from the group used?

If refer to the time series we indirectly group the information of the two or three images but is only in the database.
- Table 6 - what is the role of the "date of analysis" column, does it always coincide with the last image date?

No. It is not always the last image, the “date of analysis” is a date within the agricultural cycle where it is known with certainty that the crop has already emerged and has not been harvested

Section g, page 11:
- line 334: What does it mean that "all the phenological stages within the agricultural cycle were classified" - were separate classifiers trained for each phase, or was there a single classifier trained on all of the data?

Is only a single classifier for all data
- line 337: the term "global precision" is wrong, the authors are evaluating overall accuracy (OA); later in line 376 they call it "global accuracy", only in table 9 the term "overall accuracy" is used
- what is "precision of reference samples"? why is the acronym PMR?
- "precision of classifier" should be explained as well; also, precision metric is normally accompanied by recall and F1, because it can be trivially boosted in isolation; if PC really refers to classifier precision, recall and F1 should also be reported

We had changed the nomenclature according to English language and instead of F1 score we used Kappa index as a better indicator than overall accuracy of the whole data.

- SVM is inherently a binary classifier, it should be noted what method of multi-class methodology was used here (e.g., one-vs-rest)?

The method multi-class used was one-vs-one.

Section h, page 11, lines 365-366: "To enhance the database, the values of the pixels in the satellite images that made up each plot were averaged." - why is this process different from the one for preparing the training data?

The only difference with the training data is that when the models were validated for 2020 crop cycle we use the whole polygon of each plot.
Why does that "enhance the database"?

The redaction was changed, the mean of the statement was how the attribute table of shapefile was filled.

It should be explicitly emphasized that the results in sections 3.1-3.3 are validation results for the samples from the same season as the training data.
We have emphasized the difference between the two validations
Page 12, section 3.1:
- precision and accuracy are mixed in the text, the authors should check the definitions of accuracy and precision metrics for binary classifiers, and carefully correct all of the references to proper terms
- in Table 9, text at the bottom, is another source of ambiguity: PC is explained as classifier accuracy, before it was defined as precision of classifier
We correct the abbreviations to avoid ambiguity
Page 13, section 3.2:
- in line 404, what is PG? before it was GP, which is itself wrong, as explained before
- in lines 409-413, the "while" part of the sentence does not follow from the first part of the sentence, since it is talking about a different aspect
We change the conjunction by the adverb “instead”
Page 13, section 3.3:
- Figure 8 should in fact be Table 11, and should not be split across pages; the table structure is also messed up
- the statement in lines 435-436 is speculative and not convincing - what does the DB structure have to do with classification efficiency?
No, the statement was changed
Page 14, section 3.2:
- the section number is wrong, it should be 3.4
- the contents of tables 11-14 are unclear - what does it mean that the "combination includes image acquired on May 6" and "classification performed for the scene acquired on May 6"? As I understand, it is not the image itself that is being analyzed or classified, it is simply the last image in the input temporal sequence.

Yes the correct sentences in the title is “Results of the classification at May 6, 2020”
- Table 12 is split across pages and hard to inspect
- in line 490, accuracy is again used to refer to bean classifier "precision"

Bibliography should be thoroughly checked, e.g. in bibliography item [25] by Chakhar et al. the title is incomplete.
Bibliography was checked

Minor issues:

Table 1 is in Spanish, its caption is on a previous page.

The tables and figures were corrected to English language.

Page 2, line 81: the authors state that "the training samples result from the correlation between the field observations and the plots identified in the Sentinel-2 images" - what correlation are they talking about, and how does it define the training samples? I didn't find any mention of these correlations later on. What is the reason for periodically revisiting the plots besides determining the ground truth value for the classifier? These points is necessary to explain.

The sentences where rewrite for a better comprension

Are parcels in Figure 2 color coded in some way? This information should be added to the figure caption.

A legend chart was added to the image

Figure 3 and Table 3 are in Spanish.

The tables and figures were corrected to English language.

By substituting equation Eq. (3) into Eq. (2), we get WDVI=0, which is puzzling. I suggest to reformat the text and explain in advance that the value of a is calculated on bare land, then used on data during the crop season.

The equation was changed and emphasize the bare soil

The information in lines 298-300 on page 9 is a bit unclear without reading the caption of Figure 7. I suggest to rephrase that the threshold WDVI value for crop presence was determined based on the established state of the plots during regular visits no later than 10 days after emergence.
We accepted your suggest and it was changed
Page 10, equation (5) and (6) - I suggest using bold for vectors and italic for scalars. Making notation consistent is also required. Explanations of r and x_i in Equation (6) are missing.
The mean of parameter where named
Page 10, Equation (7): How does the equation represent a cubic polynomial? What is "(x,x_i)", and what is the value of d? What is the interpretation of addition of 1 to a pair of vectors?

Is a kernel cubic when d=3 the text was changed for a better interpretation.

Reviewer 3 Report

The paper must clarify the main goal of the crop recognition - what is the scope of this recognition. If it is about crop monitoring (as in title), information about quality of crop must be detected. Also what about presence of the weeds?

Also, the following aspects must be clarified:

-a section with related works must be added (including also existing results)

-explain how were selected the parameters of the tested methods (eg: kernel, C for SVM, etc)

-comparison with other existing methods must be extended, providing good and better results 

-depending of the scope of recognition, the inference time must be important: what is this value?

-Tables 1 and 3 must be translated in English

Author Response

RESPONSE TO REVIEWER 3

Dear Reviewer

First of all, I would like to thank you for your insightful comments, which will undoubtedly greatly improve the manuscript.

Comments and Suggestions for Authors

Responses to the referees’’ comments.

Review3

The paper must clarify the main goal of the crop recognition - what is the scope of this recognition. If it is about crop monitoring (as in title), information about quality of crop must be detected. Also what about presence of the weeds?

The objective of the work was emphasized as well as its scope. The monitoring was for the purpose of verifying the presence of the crop, mainly before and after emergence, in order to calculate the slope of the soil line used in the weighted difference vegetation index (WDVI).

-a section with related works must be added (including also existing results)

A section named “Related works was added for comparison and discussion”

-explain how were selected the parameters of the tested methods (eg: kernel, C for SVM, etc)

The parameters were tested within the classification learner tool included in Matlab, the combination that obtained the best results was chosen. These parameters were added to the document

-comparison with other existing methods must be extended, providing good and better results 

The comparison is also added in section related works and discussed later inside the document

-depending of the scope of recognition, the inference time must be important: what is this value?

The high accuracy was obtained in the middle of crop cycle using two images with a with a difference of 30 days between them

-Tables 1 and 3 must be translated in English

The tables were corrected and written in English

Reviewer 4 Report

The topic selection is meaningful. However, some content of this paper need to be further optimized. And I have some considerable concerns that would be worth addressing. I would like to give some suggestions as below.

Several problems need to be considered:

1. Maps and presentations of the study area need to be optimized for an international audience. I can't understand the study area very well because of the current statement.

2. The abstract needs to be rewritten. It doesn’t indicate the background of the research or highlight the significance of the research now.

3. Please strengthen the summary of the current research status and highlight the research significance in the introduction of the paper.

4. It is suggested to write the conclusion and discussion in two parts, and summarize the discussion into several points. This would be clearer for journal readers and highlight the innovation of the research.

5. The study appears to only have data from 2019. Are the results reliable?

6. There are too many presentations of the results so far, more like an experimental report than a research paper.

7. I suggest that the paper discuss the application potential of the research.

8. The conclusion should highlight the research innovation.

Overall, the manuscript may need to be serious modified before published.

Author Response

RESPONSE TO REVIEWER 4

Dear Reviewer

First of all, I would like to thank you for your insightful comments, which will undoubtedly greatly improve the manuscript.

Comments and Suggestions for Authors

Responses to the referees’’ comments.

Review 4

1. Maps and presentations of the study area need to be optimized for an international audience. I can't understand the study area very well because of the current statement.

The ubication map was corrected to clarify and locate better the study area.

2. The abstract needs to be rewritten. It doesn’t indicate the background of the research or highlight the significance of the research now.

The abstract was modified and emphasize this two aspects.

3. Please strengthen the summary of the current research status and highlight the research significance in the introduction of the paper.

We add a section of contributions of this work in the abstract and introduction

4. It is suggested to write the conclusion and discussion in two parts, and summarize the discussion into several points. This would be clearer for journal readers and highlight the innovation of the research.

We don’t separate the part but aggregate a better discussion at this apart

5. The study appears to only have data from 2019. Are the results reliable?

Studies have been carried out using more years, but with statistics data and without verifying in field. However, the strength of this work is the validation of training data and results in field, that make the results reliable. Despite this, a greater number of years would improve the models.

6. There are too many presentations of the results so far, more like an experimental report than a research paper.

We add a figure with the general methodology for justification of many results

7. I suggest that the paper discuss the application potential of the research.

We discus in deep the application of the research

8. The conclusion should highlight the research innovation.

We rewrite some sentences of the conclusion for the emphasize de innovation

Round 2

Reviewer 1 Report

Most of my comments are addressed. I recommend acceptance of this manuscript.

Author Response

I would like to thank you for your collaboration, rest assured that the manuscript was greatly improved.

Greetings

Reviewer 2 Report

Review of paper "Monitoring and identification of agricultural crops through multitemporal analysis of optical images and machine learning algorithms"

The revised paper answers most of my previous comments, but some issues remain.

The paper is an interesting report of a crop classification experiment, and I can appreciate the work invested, but for a research paper the original contributions of the study are not well explained. If the paper is revised again, I suggest adding a clear-text alongside the text with visible corrections.

The abstract contains many grammar errors, there was obviously no proofreading involved. Some sentences are too long (e.g. lines 19-25) and should be split into 2 or even more. The word "In" at the beginning should be removed.

Line 104: Not all bands are 20m, B2-4 are 10m. There is also an "y" instead of "and" in the parentheses.

The related work section also contains errors:
- "use ML classification methods" is missing an "of"
- the sentences in line 134 are not properly separated
- in line 136 "between them" should be ", among others"
- "ground true" should be "ground truth"
- Table 1 uses Spanish in the last column. I suggest explaining the acronyms in the table caption.

Regarding the contributions - it is still not clear that any methodological novelty is being proposed:
- according to Table 1, some previous approaches perform "within season mapping", so how is that extended by "mapping throughout agricultural cycle" used in the proposed approach.
- why is using the built model in the following agricultural cycles a particular contribution of the paper? Being able to use the model for future automation is the reason for building the model in the first place, so the authors should explain why they think their approach is different from existing ones in this respect.
- I assume other researchers had to deal with cloud occlusion in some way when building their models, so I would expect a better explanation of how exactly the proposed approach is different or why this is considered an original contribution.
Related to this, the following question occurs: how is the model to be used for future prediction, if the emergence of crops needs to be determined by visiting the plot?

Line 315: The sentence should start with "Here,", n should be italicized, and dot added at the end.

In Table 10, it is not necessary to explain the meaning of PA and UA again, while OA can be used for overall accuracy, since all of the acronyms have been previously introduced (and are also fairly standard). The same comment applies to Tables 13-16.

Line 411: Vector notation was changed to bold in Eq. (5), but inconsistently (check F(x)), but not in the text. The bias b should be italic.
Equation 6 is still not consistent with this formatting. The notation "(x,x_i)" in Equation 7 is still not corrected, I believe it should be dot product. There is some repetition of text in lines 421 and 426-427.

Line 420: Spanish "y" is used for "and", I assume. Vectors x and x_i are italic again.

Line 527: replacing while by instead did not improve the situation, I suggest splitting the sentences.

Table 12 caption contains errors. Global accuracy is used in the table, should probably be OA.

The authors replied to my comment regarding the captions of Tables 13-16 by "Yes the correct sentences in the title is “Results of the classification at May 6, 2020”", but the captions in the revised paper again talk about "combinations that included the image acquired on".

Figure 11: Kappa coefficient is denoted by K, the use of Greek letter kappa would be preferred. In the Figure itself, I suggest changing the axis labels to kappa and OA sa well. The caption can be shortened by removing the explanation of how many weeks is 15 days or 30 days.

Since there is no clear-text provided for the revised paper, it is hard to detect many formatting and grammar errors (especially missing punctuation marks).

Author Response

Cover letter for Review 2

 Dear Reviewer

 First of all, I would like to thank you for your insightful comments, which will undoubtedly greatly improve the manuscript.

 The following is an explanation of how they were treated.

The revised paper answers most of my previous comments, but some issues remain.

The paper is an interesting report of a crop classification experiment, and I can appreciate the work invested, but for a research paper the original contributions of the study are not well explained. If the paper is revised again, I suggest adding a clear-text alongside the text with visible corrections.

The abstract contains many grammar errors, there was obviously no proofreading involved. Some sentences are too long (e.g. lines 19-25) and should be split into 2 or even more. The word "In" at the beginning should be removed.

The abstract was revised and some sentences were rewrite for reduce them. 

Line 104: Not all bands are 20m, B2-4 are 10m. There is also an "y" instead of "and" in the parentheses.
the redaction was changed and explain that the 10 m bands were resampled to 20 m.

The related work section also contains errors:
- "use ML classification methods" is missing an "of"
- the sentences in line 134 are not properly separated
- in line 136 "between them" should be ", among others"
- "ground true" should be "ground truth"
- Table 1 uses Spanish in the last column. I suggest explaining the acronyms in the table caption.

The errors where checked and corrected and the list of acronyms added.

Regarding the contributions - it is still not clear that any methodological novelty is being proposed:
- according to Table 1, some previous approaches perform "within season mapping", so how is that extended by "mapping throughout agricultural cycle" used in the proposed approach.

Most of them research use historical statistics of ground information for generate cropland maps but not always is possible to have that information, therefore, this proposal is better.

- why is using the built model in the following agricultural cycles a particular contribution of the paper? Being able to use the model for future automation is the reason for building the model in the first place, so the authors should explain why they think their approach is different from existing ones in this respect.

We add a figure for to try explain in a better way the construction of models and we think that explain why is different of previous works.

- I assume other researchers had to deal with cloud occlusion in some way when building their models, so I would expect a better explanation of how exactly the proposed approach is different or why this is considered an original contribution.

The figured added explain how we deal with the cloud presence in the images, and how more than one pattern for each plot is considered in the model during the crop cycle.

Related to this, the following question occurs: how is the model to be used for future prediction, if the emergence of crops needs to be determined by visiting the plot?

No, the emergence is inferred with the value of WDVI, this should be major than 0.005. A value of 0 or minor in WDVI indicate the absence of vegetation, but is necessary the field surveys before the emergency of crops (bare soil) and calculate the slope of soil, necessary for the WDVI index.

Line 315: The sentence should start with "Here,", n should be italicized, and dot added at the end.

The sentences was corrected.

In Table 10, it is not necessary to explain the meaning of PA and UA again, while OA can be used for overall accuracy, since all of the acronyms have been previously introduced (and are also fairly standard). The same comment applies to Tables 13-16.

The meaning of acronyms were removed

Line 411: Vector notation was changed to bold in Eq. (5), but inconsistently (check F(x)), but not in the text. The bias b should be italic.
Equation 6 is still not consistent with this formatting. The notation "(x,x_i)" in Equation 7 is still not corrected, I believe it should be dot product. There is some repetition of text in lines 421 and 426-427.
Line 420: Spanish "y" is used for "and", I assume. Vectors x and x_i are italic again.

Equations 5, 6 and 7 were corrected

Line 527: replacing while by instead did not improve the situation, I suggest splitting the sentences.

The sentences was rewrite for a better understanding

Table 12 caption contains errors. Global accuracy is used in the table, should probably be OA.

The error was corrected

The authors replied to my comment regarding the captions of Tables 13-16 by "Yes the correct sentences in the title is “Results of the classification at May 6, 2020”", but the captions in the revised paper again talk about "combinations that included the image acquired on".

We had correctly changed the titles this time.

Figure 11: Kappa coefficient is denoted by K, the use of Greek letter kappa would be preferred. In the Figure itself, I suggest changing the axis labels to kappa and OA sa well. The caption can be shortened by removing the explanation of how many weeks is 15 days or 30 days.

The title of figure and axis were reduced.

Since there is no clear-text provided for the revised paper, it is hard to detect many formatting and grammar errors (especially missing punctuation marks).

We checked the formatting and grammar to try find the errors and corrected

Reviewer 3 Report

Since all my comments were addressed, I recommend to publish the paper.

Author Response

(The authors gave the same response as above.)

Reviewer 4 Report

The authors have carefully revised the manuscript, but there are still some problems, please continue to improve the manuscript.

1. Writing the conclusion and discussion in one section are not conducive to reading, please further summarize the content.

2. The paper has little discussion on the limitations and future directions of the manuscript.

3. Some details can be improved. For example, the text in Figures 1 and 2 is so small, and Tables 3 and 9 can be expressed in words.

I suggest further refinement of the manuscript.

Author Response

Cover letter for Review 4

 Dear Reviewer

 First of all, I would like to thank you for your insightful comments, which will undoubtedly greatly improve the manuscript.

 The following is an explanation of how they were treated.

Comments and Suggestions for Authors

Responses to the referees’’ comments.

1. Writing the conclusion and discussion in one section are not conducive to reading, please further summarize the content.

Sorry the sections that are together and mentioned in the previous document was the results and discussion. The discussions and conclusions are separated in the document. We checked the discussions and enriched them and we checked and reduced the conclusion

2. The paper has little discussion on the limitations and future directions of the manuscript.

We include a better discussion about the limitations and future directions

3. Some details can be improved. For example, the text in Figures 1 and 2 is so small, and Tables 3 and 9 can be expressed in words.

The tables mentioned were removed so table 8 and the figures was improvement.

I suggest further refinement of the manuscript.

We checked the formatting and grammar and we corrected some errors
